# Genome-Wide Identification of the NPR1-like Gene Family in *Solanum tuberosum* and Functional Characterization of *StNPR1* in Resistance to *Ralstonia solanacearum*

**DOI:** 10.3390/genes14061170

**Published:** 2023-05-27

**Authors:** Fumeng He, Dexing Kong, Zhe Feng, Yongqing Xu, Qiang Yuan, Dan Liu, Xue Wang, Xu Feng, Fenglan Li

**Affiliations:** 1College of Life Sciences, Northeast Agricultural University, Harbin 150030, China; hefumeng@neau.edu.cn (F.H.); 13774979240@163.com (D.K.); yuti8221@163.com (Y.X.); neauyqiang@sohu.com (Q.Y.); liudan1533@163.com (D.L.); wang_x2022@126.com (X.W.); 18045043687@163.com (X.F.); 2Pharmacology & Toxicology Department, Saint Joseph’s University Philadelphia College of Pharmacy, Philadelphia, PA 19104, USA; 13163443020@163.com

**Keywords:** *Solanum tuberosum*, *Ralstonia solanacearum*, *NPR1*-like gene, disease resistance, systemic acquired resistance

## Abstract

The NPR1 (nonexpressor of pathogenesis-related genes 1) gene is an activator of the systemic acquisition of resistance (SAR) in plants and is one of the central factors in their response to pathogenic bacterial infestation, playing an important role in plant disease resistance. Potato (*Solanum tuberosum*) is a crucial non-grain crop that has been extensively studied. However, the identification and analysis of the *NPR1*-like gene within potato have not been understood well. In this study, a total of six NPR1-like proteins were identified in potato, and phylogenetic analysis showed that the six NPR1-like proteins in *Solanum tuberosum* could be divided into three major groups with NPR1-related proteins from *Arabidopsis thaliana* and other plants. Analysis of the exon–intron patterns and protein domains of the six *NPR1*-like genes from potato showed that the exon–intron patterns and protein domains of the NPR1-like genes belonging to the same *Arabidopsis thaliana* subfamily were similar. By performing quantitative real-time PCR (qRT-PCR) analysis, we found that six NPR1-like proteins have different expression patterns in different potato tissues. In addition, the expression of three *StNPR1* genes was significantly downregulated after being infected by *Ralstonia solanacearum* (RS), while the difference in the expression of *StNPR2/3* was insignificant. We also established potato *StNPR1* overexpression lines that showed a significantly increased resistance to *R. solanacearum* and elevated activities of chitinase, β-1,3-glucanase, and phenylalanine deaminase. Increased peroxidase (POD), superoxide dismutase (SOD), and catalase (CAT) activities, as well as decreased hydrogen peroxide, regulated the dynamic balance of reactive oxygen species (ROS) in the *StNPR1* overexpression lines. The transgenic plants activated the expression of the genes associated with the Salicylic acid (SA) defense response but suppressed the expression of the genes associated with Jasmonic acid (JA) signaling. This resulted in resistance to *Ralstonia solanacearum*.

## 1. Introduction

Potato is a crop of the Solanum genus and is considered a vegetable and food crop in many countries because of its high nutritional content [1]. In recent years, China has become one of the leading potato producers in the world in terms of its production and planting area [2]. Bacterial wilt (BW) caused by *Ralstonia solanacearum* (RS) is one of the major diseases that occurs in potato production, and can cause massive yield losses in potato, up to 80% in severe cases [3]. *R. solanacearum* is a pathogen that damages the vascular tissue system of plants by invading the root xylem and spreading to the stem tissue, whereby excess extracellular polysaccharides and a variety of extracellular proteins that can block and destroy the ducts are produced, causing the plant to lose water and wilt [4]. Nowadays, the use of resistant varieties is the most cost-effective way to prevent *R. solanacearum* in potato production. Therefore, screening potato germplasm resources for resistance to *R. solanacearum* plays a key role in potato disease breeding [5].

*NPR1* is a key gene in the systemic acquired disease resistance response of plants and plays an important role in the regulation of broad-spectrum resistance in plants. The NPR1 protein contains an N-terminal BTB/POZ (broad-complex, tramtrack, and bric-a-brac/pox virus and zinc finger) structural domain and an anchor protein repeat domain ANK (ankyin repeat domain) located in the middle region. These two domains play important roles in protein interactions [6]. At present, *NPR1* genes and homologs have been isolated from a variety of plants, including *Arabidopsis thaliana* [7], *Oryza sativa* [8], *Triticum aestivum* [9], *Malus domestica* [10], and *Anemone vitifolia* [11], and the functions of some of them have been investigated to different degrees. Phylogenetic analysis separates the NPR1-like proteins into three distinct clades, suggesting functional divergence [12]. *AtNPR1* and *AtNPR2*, in the first branch, are essential receptors for salicylate metabolism and act as transcriptional co-activators in plant immunity [13,14]; *AtNPR3* and *AtNPR4*, in the second branch, are also important receptors for salicylate metabolism and act as transcriptional co-repressors in plant defense [14,15]. The *AtNPR5* and *AtNPR6* genes, also known as *AtBOP1* and *AtBOP2*, respectively, which lack the NPR1-like C-terminal structural domain, are involved in plant growth and development, especially in the formation of floral organs [16,17].

Plants have developed various immune defense mechanisms to resist several biotic and abiotic stresses during the organism’s evolution [18]. There are two categories of systemic resistance mechanisms that occur in plants; one is systemic acquired resistance (SAR) induced by pathogenic bacteria, and the other is induced systemic resistance (ISR) induced by non-pathogenic microorganisms [19]. The *NPR1* gene is a crucial regulator of the development of broad-spectrum disease resistance in plants, which mainly regulates the expression of disease-process-related proteins and plays a key role in regulating the development of SAR. The *NPR1* gene also has an important function in the regulation of the Jasmonic acid (JA)-mediated ISR response, which is a negative regulator of the JA signaling pathway [2]. Meanwhile, *NPR1* is a vital gene in plants’ response to pathogen attacks and is involved in the balance of SA and JA in plants [20].

Recent studies have shown that NPR1 proteins bind to each other via disulfide bonds and exist as multimers in the cytoplasm when plants are not attacked by pathogens. Upon infection with pathogenic bacteria, the accumulation of SA alters the redox potential in the cell, leading to the release of NPR1 monomers, which enter the nucleus and bind to the TGA-bZIP transcription factor, thereby activating the expression of defense-response-related genes and ultimately causing a SAR response [7]. The overexpression of the *AtNPR1* gene in *Arabidopsis thaliana* enhances the resistance of the plant to *Pseudomonas syringae* and *Peronospora parasitica* [21]. Additionally, the mutation of the *NPR1* gene makes the plant more susceptible to these pathogens [22].

Potatoes are susceptible to pathogenic infections during growth due to their environment and other factors, which can seriously affect their yield and quality. Therefore, disease control during potato growth is crucial. However, traditional control approaches are time-consuming and labor-intensive, leading to an elevation in the cost of cultivation. Currently, genetic engineering is a major method used in potato disease resistance breeding, which uses existing germplasm resources to breed resistant potato varieties. Molecular breeding is the most cost-effective way to control pests and diseases. In this study, the *NPR1*-like gene family of potato was identified. Phylogenetic analysis, gene structure analysis, gene chromosomal localization, and the response pattern upon infection by *R. solanacearum* were performed on *StNPR1*-like genes. In addition, we cloned the *StNPR1* gene from potato and analyzed the subcellular localization of the *StNPR1* gene. Meanwhile, using *StNPR1* gene overexpression lines, we explored the role of the *StNPR1* gene in potato resistance to *R. solanacearum* stress, which can provide a new method for the control of *R. solanacearum*.

## 2. Materials and Methods

### 2.1. Bioinformatics Analysis of StNPR1

The potato reference genome was from the Spud DB database (http://spuddb.uga.edu/, accessed on 13 October 2022). A hidden Markov model (HMM) of the NPR1 family members was built using the conserved domains of BTB/POZ (PF00651) and ankyrin repeat (PF00023) derived from the Pfam database (https://pfam.xfam.org/, accessed on 6 December 2022) and searched against the genome of potato. NPR1 family members were then screened and identified using the NCBI-CDD website. Using MEGA7.0 software, the amino acid evolutionary tree was created via the neighbor-joining process [23].

### 2.2. Plant Growth and Disease Treatment

The potato (Atlantic variety) used in this study was saved and provided by the College of Life Sciences, Northeast Agricultural University. The potato culture condition was 28 °C, and the light cycle was 16/8 h. Potato roots, stems, leaves, flowers, and tubers were collected as samples, quick-frozen in liquid nitrogen, and stored in a −80 °C refrigerator. The pathogenic strain of the bacterial treatment was the *Ralstonia solanacearum* (RS) strain PO41 (minor species 3) (biovar 2). The *R. solanacearum* bacterial suspension (OD_600_ = 0.8) was used to treat potato seedlings at the 8–9-leaf stage and inoculated using root inoculation bacteria [10]. We used water as the control. Potato leaves were sampled 0, 3, 6, 12, and 24 h after pathogen treatment, quick-frozen in liquid nitrogen, and stored in a −80 °C refrigerator for later use.

### 2.3. qRT-PCR

The transcription levels of six *StNPR1*-like genes in different potato tissues and their responses to biological stress were determined via qRT-PCR. In order to perform qRT-PCR, total RNA was extracted using Trizol according to the instructions of the plant RNA extraction kit (Takara, Dalian, China). The removal of genomic DNA contamination was performed using the Ambion Turbo DNase kit (Life Technology, Carlsbad, CA, USA). A Reverse Transcription System (Promega, Madison, WI, USA) was used to reverse transcribe the extracted total RNA to synthesize cDNA. The PCR reaction was performed using the actin gene as the internal reference gene (GenBank accession number: X55747). Then, we calculated the relative expression levels of the target gene and the internal reference gene using the 2^−∆∆CT^ method [24]. The primers used in this study are listed in Appendix A.

### 2.4. Subcellular Localization of StNPR1 Protein

Using a pretest, we selected the *StNPR1* gene—which is upregulated in response to biological stress—as the research object and obtained the full length of the *StNPR1* gene using the cDNA extracted previously via PCR. *StNPR1* was cloned into a pCambia1302::eGFP vector using the homologous recombination method to generate a STNPR1-eGFP fusion protein. The 35S::eGFP and 35S::StNPR1-eGFP were transferred into *Agrobacterium tumefaciens* LBA4404 using the liquid nitrogen freeze–thaw method. We determined the subcellular localization of StNPR1 via transient transfection into the leaf epidermal cells of 1-month-old *Nicotiana benthamiana*. Subcellular localization experiments were conducted according to Chen et al. [25]. The transformed tobacco leaf epidermal cells were observed via confocal microscopy.

### 2.5. Overexpression of StNPR1 in Potato

The 35S::STNPR1-GUS vector was constructed via homologous recombination and by connecting *StNPR1* to the PBI121 vector. The T-DNA region of the vector contained a Kanamycin sulfate resistance gene, which could be used for screening transgenic plants. The 35S::StNPR1-GUS vector was transformed into *Agrobacterium tumefaciens* LBA4404 via the freeze–thawing process. The micropotato genetic transformation system was used to cut the micropotato (0.5–1.0 cm in diameter) into slices approximately 2 mm thick. The potato tuber slices were infected with the above *Agrobacterium suspension* (OD_600_ = 0.5) for 7 min. The infected potato tuber slices were co-cultured in MS medium (1 mg/L of 6-Benzylaminopurine + 1.5 mg/L of Indole-acetic acid + 50 μmol/L of Acetosyringone) at 27 °C in the dark for 2 days. The potato tuber slices were then transferred to MS differentiation medium (1.0 mg/L of 6-Benzylaminopurine + 1.0 mg/L of Indole-acetic acid + 2 mg/L of Zeatin + 50 mg/L of Kanamycin + 250 mg/L of Ceftiofur). When the regenerated seedlings grew to 1–2 cm, they were placed in MS medium (50 mg/L Chlormequat chloride + 75 mg/L Kanamycin + 250 mg/L Ceftiofur) for rooting selection in order to obtain transgenic plants. GUS gene-specific primers were used for PCR identification. The expression of the StNPR1 gene in transgenic potato was detected using RT-qPCR. The process of RNA extraction and cDNA synthesis is described above.

### 2.6. Analysis of Resistance of StNPR1 Transgenic Plants to R. solanacearum

The *StNPR1*-overexpressed potato and wild-type potato were placed in a greenhouse for 60 days. Potato leaves were cultured in vitro at 28 °C with a constant temperature, and the light cycle was 16/8 h. Isolated potato leaves were inoculated with 20 uL of the *R. solanacearum* bacterial suspension (OD_600_ = 0.8) to spread the bacteria into the veins. The disease symptoms in the isolated leaves of the overexpressed and wild-type plants were compared between days 1 and 4 after inoculation with the pathogen [10]. The apoptotic cells affected by bacteria were observed using the trypan blue staining method, and samples were taken at the same time [26]. Then, the cells were quickly frozen in liquid nitrogen and stored in the refrigerator at −80 °C for use.

In order to explore the defense response mediated by the *StNPR1* gene, we selected four genes associated with the SA defense response, including the *PR1* downstream transcription factor *WRKY70*, the disease-course-associated gene *PR1*, the isochorismate synthase gene *ICS1*, and the Phytoalexin-deficient 4 gene *PDA4* [27,28,29,30]. Transcriptional analysis was also performed on the JA biosynthesis gene *AOS*, JA response gene VSP1, and the marker gene *PDF1.2,* which is associated with the JA defense response [31,32,33]. The qRT-PCR method was the same as above. The primers used in this study are listed in Appendix A. The defense-related β-1, 3-glucanase (Glu), chitinase (CHI), phenylalanine ammonia lyase (PAL), and polyphenol oxidase (PPO) activities were determined [34]. The antioxidant-related superoxide dismutase (SOD), peroxidase (POD) and catalase (CAT) (Nanjing Jiengcheng) activity, and the hydrogen peroxide (H_2_O_2_) content (Suzhou Keming), were also measured [35].

### 2.7. Statistical Analysis

The expression analysis data were statistically analyzed for three biological replicates and three technical replicates, and the data are shown as the mean ± standard deviation. Statistical analysis was performed using SPSS 20.0. The difference was statistically significant with a probability (*p*) value ≤ 0.05, as determined by *t*-test.

## 3. Results and Analysis

### 3.1. Identification, Phylogeny, and Characterization of Potato NPR1-like Genes

The conserved BTB/POZ (PF00651) and ankyrin repeat (PF00023) structural domains of the NPR1 family members were identified via Blast search and HMMER analysis in the potato genome database and then screened by using pfam and the NCBI-CDD website to remove the duplicate sequences and structural domain variants. A total of six NPR1 family members were identified in potato and named *StNPR1-6* (Appendix A). The amino acid sequence encoded by *StNPR4* was the shortest (461aa), and the amino acid sequence encoded by *StNPR1* was the longest (627aa). The molecular weight of the NPR1 family ranged from 50,646.89 kDa to 69,593.25 kDa, while the theoretical isoelectric point ranged from 5.66 to 6.46, indicating that the proteins encoded by these six members were weakly acidic. The SPUD DB website was used to analyze the chromosome location of the potato *NPR1* gene family members, and two genes were found on chromosome 7, namely *StNPR1* and *StNPR2*. *StNPR3* was distributed on chromosome 2, and the *StNPR4* gene was distributed on chromosome 4. Meanwhile, *StNPR5* and *StNPR6* were distributed on chromosome 10 (Appendix A).

We analyzed the evolutionary relationships of the NPR1 family members, and the results showed that the NPR1 family members could be classified into three subclasses (I, II, and III). Among them, subclass I’s yellow area included StNPR1, *Arabidopsis thaliana* NPR1/2, *Nicotiana tabacum* NPR1, *Ipomoea batatas* NPR1, *Solanum lycopersicum* NPR1, and other proteins, indicating that they are evolutionarily close and may have a common origin. Additionally, *AtNPR1* was shown to play an important role in improving plant disease resistance [36]. In subclass Ⅱ’s dark-red area, StNPR2 and StNPR3 were in the same branch with AtNPR3/4, *Oryza sativa* NPR3, *Zea mays* NPR1, and other proteins, suggesting that the *StNPR2* and *StNPR3* genes may be involved in the negative regulation of SAR [37]. In branch Ⅲ’s light-blue area, StNPR4, StNPR5, and StNPR6, three members of potato, were in the same branch as AtNPR5/6, NtBOP3/4, and OsNPR5. Additionally, it was speculated that these seven genes may be related to plant organ development [16] (Figure 1). Overall, the *NPR1* gene was relatively conserved during evolution.

### 3.2. Sequence and Structural Analysis of StNPR1-like Genes and Proteins

To further explore the potential functions of the *StNPR1*-like gene family, the structural features and nucleotide sequences of the six potato NPR genes were systematically analyzed. The number of introns in this family was 1–4, among which, *StNPR1* was divided into five segments by four introns. Both *StNPR2* and *StNPR3* were divided into four fragments by three introns. *StNPR4*, *StNPR5*, and *StNPR6* consisted of one intron and two exons (Figure 2A).

In order to further clarify the biological conservation of the potato StNPR protein motifs, domains, amino acid residue mutation sites, and protein functional residues, multiple sequence alignment was performed on the six StNPRs and *Arabidopsis thaliana* (AtNPR1 to AtNPR6), *Oryza sativa* (OsNPR1), and *Nicotiana tabacum* (NtNPR1) NPRs with defined function protein sequences (Figure 2B). The amino acid sequence mutation sites npr1-1 (His334Tyr), npr1-2 (Cys150tyr), and nim1-2 (His300tyr) of the Arabidopsis AtNPR1 allele mutants were highly conserved among the six StNPR, OsNPR1, and NtNPR1 proteins. The three cysteine residues (Cys82, Cys156, and Cys216) associated with oligomer formation in the AtNPR1 protein and the cysteine residues Cys82, Cys156, and Cys216 in the six StNPRs proteins were highly conserved. In addition, the AtNPR3/4 proteins contained the EAR motif (VDLNETP), which is essential for their transcriptional repression [14]. This transcriptional repressor motif is also intact in the StNPR3 protein. In conclusion, Arabidopsis and potato NPR1-like family members that are clustered in the same evolutionary branch share similar gene structures, conserved structural domain types, and positional arrangements, and are also highly conserved at key amino acid sites; presumably, their homologous genes have similar biological functions in potato.

### 3.3. Expression Analysis of StNPR1-like Genes in Different Tissues and Organs

In order to investigate the tissue-specific expression of the *StNPR1*-like gene members in potatoes, the expression patterns were analyzed via qRT-PCR in different potato tissues (roots, stems, leaves, flowers, and tubers) (Figure 3). The results showed that the *StNPR1*-like genes were expressed in potato roots, above-ground stems, leaves, flowers, and tubers, with significant differences in expression in different organs. *StNPR1* was expressed at moderate levels, with higher expression levels in the leaf and stolon tissues. *StNPR2* had the greatest expression level in the leaves and tubers. *StNPR3* was highly expressed in the leaves, followed by tubers. *StNPR4* had the highest expression level in the flowers. The expression of *StNPR5* in the tubers was significantly higher than that of other genes in this family. *StNPR6* showed strong expression levels in the stems, leaves, and tubers, but minor expression in other organs. 

### 3.4. Expression Analysis of StNPR1-like Genes under Biotic Stress

To investigate the potential functions of the *StNPR1*-like genes in response to biological stress, the pathogenic bacterium *R. solanacearum* was used to treat potatoes, and the expression of the *StNPR1*-like genes was detected via qRT-PCR after pathogen stress (Figure 4). The expression of the *StNPR1* gene increased first and then decreased after treatment with *R. solanacearum*, and significantly different expression levels were observed between 3 and 12 h. The *StNPR2* gene was significantly expressed at 6 h. The *StNPR3* gene was only differentially expressed at 24 h. The expression of the *StNPR4* gene first decreased and then increased, and the expression was only significantly different at 6 h. The expression of *StNPR5* and *StNPR6* genes showed a downward trend, and significant differences in the *StNPR5* gene expression were observed between 3 h and 24 h. At 12 h, no statistically significant differences were observed in the expression levels of four genes, namely *StNPR2*, *StNPR 3*, *StNPR4,* and *StNPR 6*, while *StNPR1* was significantly upregulated. Taken together, the *StNPR1* gene, upregulated in response to pathogenic bacterial stress, was selected as the next research object to verify its biological function.

### 3.5. Subcellular Localization of StNPR1 Gene in Tobacco

Sequence analysis showed that StNPR1 contains a nuclear localization signal (NLS) at the C-terminal of the StNPR1 amino acid sequence, and that the STNPR1 protein may be localized in the nucleus. We performed the subcellular localization of the StNPR1 protein using *Nicotiana benthamiana* leaves as a model and then observed it under a fluorescent microscope. The StNPR1-EGFP fusion protein was localized in the nucleus, while the EGFP, as the control protein, showed fluorescence throughout the cells. These results indicate that the StNPR1 protein is localized in the nucleus (Figure 5).

### 3.6. Overexpression of the StNPR1 Gene Enhances Potato Resistance to R. solanacearum

To further investigate the function of the *StNPR1* gene in potato disease resistance, two *StNPR1* transgenic plants were obtained using the *Agrobacterium tumefaciens*-infected potatoes micropotato transformation method. Further analysis was conducted on overexpressed 1 (OE-1) and overexpressed 3 (OE-3) plants (Figure 6B). The isolated leaves of transgenic plants were inoculated with the pathogen *R. solanacearum* to observe and measure the disease spots. The results showed that the lesions on OE-1 and OE-3 leaves were significantly smaller than those on the wild type (Figure 6C). Trypan blue dye was used to stain the leaves of the overexpressed potato, and the wild-type potato was inoculated with *R. solanacearum* to observe the cell death of different plants. The number of cell deaths in the leaves of both the OE-1 and OE-3 overexpression lines was significantly lower than that of WT at different times after *R. solanacearum* treatment, which was consistent with the spot area results (Figure 6D,E). The differences in the disease symptoms between the leaves of the transgenic and wild-type plants were further confirmed.

### 3.7. Effect of StNPR1 Overexpression on Defensive Enzyme Activities

To further analyze the enhanced resistance of the StNPR1-overexpression potato to *R. solanacearum*, the activities of the defense enzymes CHI, GLU, PPO, and PAL in transgenic and wild-type plants were analyzed 0, 1, 2, 3, and 4 dpi. The results showed that the CHI activity of the OE-1 and OE-3 overexpression lines was significantly higher than that of the WT at 2, 3, and 4 dpi (Figure 7). The GLU activity of OE1 and OE-3 was significantly higher than that of the WT, and the GLU activity of OE-1 and OE-3 reached a peak at 3 dpi. The PAL activity of the overexpression lines was also significantly higher than that of WT, and their PAL activity showed an ascending–descending–ascending–descending trend. The PPO enzyme activity in the overexpression lines increased rapidly to a peak at 2 dpi and then gradually decreased. Additionally, the difference in PPO activity between the overexpression lines and wild-type plants was significant. In conclusion, the overexpression of the *StNPR1* gene enhanced the activities of CHI, GLU, PPO, and PAL, which may be related to potato’s resistance to *R. solanacearum*.

### 3.8. Effect of Overexpression of StNPR1 on ROS Content and Antioxidant Enzyme Activity

In order to explore the effect of *StNPR1* overexpression on the content of reactive oxygen species and the activities of antioxidant enzymes in potato plants, the H_2_O_2_ content and the activities of SOD, POD, and CAT in leaves inoculated with *R. solanacearum* were measured. As shown, the H_2_O_2_ accumulation in transgenic plants was significantly lower compared to wild-type plants after inoculation with the pathogen (Figure 8). The CAT and SOD activities of both the overexpression and wild-type lines showed a trend of increasing and then decreasing, with a peak at 2 dpi, and the CAT and SOD activities of the transgenic lines were significantly higher than those of the wild-type plants. Compared with the wild-type plants, the POD enzyme activity of the transgenic plants was higher, and the difference was significant. In summary, the results showed that the transgenic plants had lower levels of reactive oxygen species accumulation and higher SOD, CAT, and POD enzyme activities than the wild-type plants under the same pathogen infection conditions.

### 3.9. Effect of Transgenic Potato Overexpression of StNPR1 on SA/JA-Signaling-Related Genes

To investigate the *StNPR1*-gene-mediated defense response, a transcriptional analysis of the SA and JA signaling-pathway-related genes was performed on transgenic lines and wild-type plants after pathogen infection. The results showed that the transcription levels of *StPR1*, the *NPR1* downstream transcription factor *StWRKY70,* and the phenylalanine ammonia-lyase gene *StPAL* were significantly increased in the OE-1 and OE-3 transgenic plants compared with the WT plants. In contrast, the expression levels of *StICS1* and *StPAD4* were not significantly different in the transgenic lines and WT plants. The JA biosynthetic gene *AOS* and the JA response gene *VSP1* showed similar expression patterns to *BnPDF1.2*. These results suggest that the overexpression of the *StNPR1* gene activates the expression of SA defense genes and suppresses the JA defense response after *R. solanacearum* infection (Figure 9).

## 4. Discussion

The *NPR1* gene plays a central regulatory role in plant disease resistance. It can improve the ability of plants to resist a variety of diseases, and it is a cross-site between a variety of disease resistance pathways in plants [6]. Members of the *NPR1* family have been isolated and identified in an increasing number of plants, with six members identified in *Arabidopsis thaliana*, four in rice, nine in poplar [38], and five in *Persea americana* [39]. In this study, we identified six *NPR1*-like genes in the potato genome and constructed a phylogenetic tree of NPR domains in potato, poplar, Arabidopsis, and rice. The NPR genes of potato were distributed in the three branches and were clearly divided into three subclasses with the six NPR1 family members of *Arabidopsis thaliana*, indicating that the potato gene family members may have a clear division of function.

The expression patterns of *NPR1*-like genes in different tissues have been reported in Arabidopsis [16] and wheat [40], but their tissue expression characteristics in potato have been rarely studied. In this study, the expression pattern of the *NPR1*-like gene was analyzed, and the results showed that the expression of the *NPR1*-like gene was tissue-specific. The expression of the *NPR1*-like gene was specific in *StNPR3* in tubers and *StNPR4* in flowers. These results indicate that they may play a special role in the growth and development of some organs, which is consistent with the tissue expression specificity of *Arabidopsis thaliana NPR1*-like genes [41]. The expression levels of *StNPR4* and *StNPR6* were similar in stems, leaves, and tubers, but minor in other organs. The results imply that *StNPR4* and *StNPR6* might have relevance to *AtBop1/2* in plant growth and development [16,42].

Bacterial wilt induced by *R. solanacearum* is one of the main diseases of potato and is the most serious bacterial disease that seriously threatens the yield and quality of potato, but no effective control measures have been found yet [43]. In this study, the expression profile of the *StNPR1*-like gene was analyzed under treatment with *R. solanacearum*, and the expression of *StNPR1* was significantly elevated after *R. solanacearum* treatment, while *StNPR2/3* was not significantly changed under the disease stress. The downregulation of *StNPR4/5/6* expression may be negatively regulated during infection, indicating that the *StNPR1* gene may play an important role in *R. solanacearum’s* defense. It has been reported that the constitutive expression of the *AtNPR1* gene can induce disease resistance in rice, and the overexpression of the *NPR1* gene in *Arabidopsis thaliana* and rice can improve their resistance to bacteria and fungi, indicating that *NPR1* plays an extremely important role in the plant defense system. In this study, the *StNPR1* gene was overexpressed in potato, and the transgenic plants showed a delayed onset of disease, a smaller lesion area, and a reduced number of dead cells. These results indicated that the overexpression of *StNPR1* could improve the resistance of potato to *R. solanacearum*.

In the process of pathogen infection, the burst of ROS, mainly hydrogen peroxide and the superoxide radical anion, is related to the HR hypersensitivity reaction, which affects the acquisition of SAR. SOD, CAT, and POD play important roles in removing excessive ROS and oxidizing substances in cells. In this study, the H_2_O_2_ concentration and the activities of CAT, SOD, and POD were determined in NPR1-overexpressed plants and WT plants. The content of H_2_O_2_ in overexpressed plants was lower than that in wild-type plants, but the activities of SOD, CAT, and POD in overexpressed plants were higher than those in wild-type plants. CAT is an H_2_O_2_ scavenger and plays an important role in H_2_O_2_ scavenging [44], which resulted in a lower H_2_O_2_ content in the overexpression plants compared to the WT plants. The excess of ROS and oxidized substances is unfavorable to plant growth. The increased activities of SOD, CAT, and POD help eliminate the excess production of ROS and oxidized substances, reduce oxidative stress during *R. solanacearum* infection, and maintain a dynamic balance that can not only produce ROS in order to resist exogenous infection, but also remove the excess ROS that will cause damage.

Salicylic acid and jasmonic acid mediate the basic signaling pathways of plant disease resistance and defense response. The SA signal transduction pathway is closely related to the JA signal transduction pathway, and the two pathways cooperate with each other and cross each other. Meanwhile, *NPR1* is an important regulatory element at the intersection of multiple defense pathways. It can function as a receptor of SA and a positive regulator of SAR, which plays an important role in SA/JA signaling. Previous studies have shown that the SA signal transduction pathway has antagonistic effects with the JA and ET-mediated signal transduction pathways. The accumulation of SA can inhibit the expression of JA-regulated genes, and the occurrence of this antagonism also requires the participation of *NPR1* [45]. The results of this study showed that the expression of disease-resistant genes related to the SA signal transduction pathway was enhanced in *StNPR1*-OE plants under *R. solanacearum* infection, and that the overexpression of *StNPR1* led to the suppression of the expression of genes related to the JA signal pathway. *StNPR1* positively regulated the SA defense response in the interaction between potato and *R. solanacearum*. In addition, NPR1 has been shown to inhibit JA-dependent defense responses in *Arabidopsis thaliana*. 

## 5. Conclusions

In summary, this is the first time that six *NPR1*-like genes have been identified in potato and *StNPR1*-like genes have been analyzed in depth, including their molecular characteristics, chromosomal distribution, phylogenetic classification, gene structure, and amino acid residues. The expression of *StNPR1*-like genes was tissue-expression-specific. *StNPR1* was significantly induced by *R. solanacearum* biological stress conditions. The StNPR1 protein subcellularly localized to the cytoplasm and depolymerized into monomers in the nucleus after exposed to pathogenic bacteria or SA treatment. Transgenic potatoes overexpressing *StNPR1* showed a significantly enhanced resistance to *R. solanacearum*, possibly related to ROS accumulation and the SA-mediated defense response. Therefore, it is possible that *StNPR1* could be an important candidate gene for genetic engineering in order to improve disease resistance.

## Figures and Tables

**Figure 1 genes-14-01170-f001:**
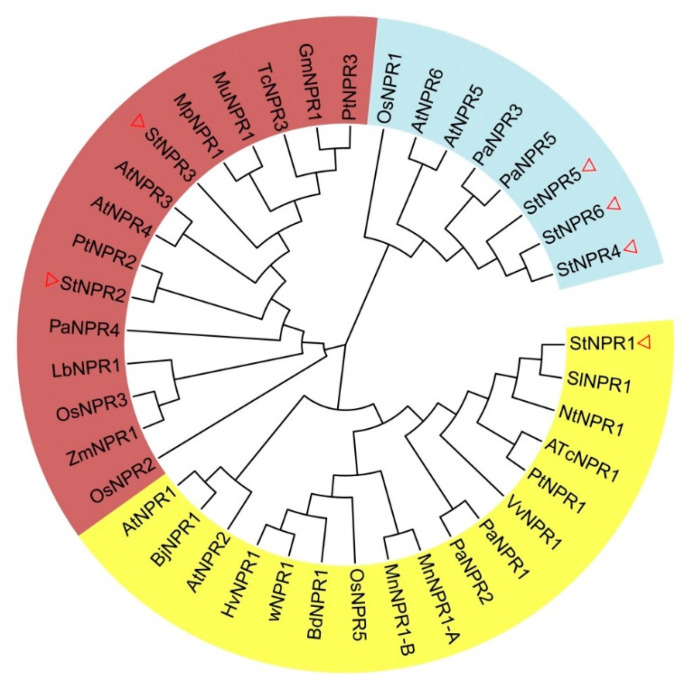
Amino acid phylogenetic tree of potato NPR1 family members. Constructed using the neighbor-joining (NJ) algorithm via MEGA7.0 software with 1000 Bootstrap replicates; the GenBank accession numbers of all species are listed in the Appendix A; and the three evolutionary branches are distinguished by different colors. The red triangle represents the position of the potato NPR1-related proteins in the three branches of the phylogenetic tree.

**Figure 2 genes-14-01170-f002:**
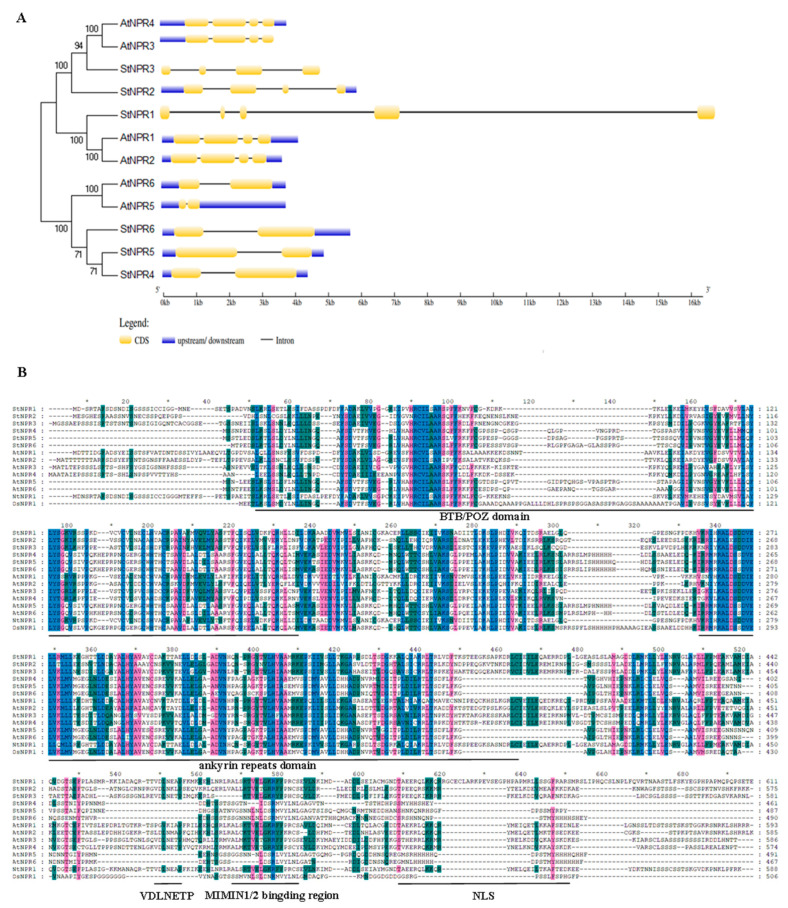
Exon–intron structure of the StNPR1-like gene. (**A**) Exons are shown as yellow boxes, UTRs are shown as blue boxes, and introns are shown as thin lines. The size of exons and introns can be estimated using the scale below. (**B**) Multiplex alignment of amino acid sequences of *Solanum tuberosum* NPR1-like proteins (StNPR1 to StNPR6) and functionally related *Arabidopsis thaliana* (AtNPR1 to AtNPR6), *Oryza sativa* (OsNPR1), and *Nicotiana tabacum* (NtNPR1) NPR1 proteins. The BTB/POZ and ANK conserved structural domains and significant motifs (LENRV), ear-like inhibitory motifs (VDLNETP), NIMIN binding region, and nuclear localization signal (NLS) are highlighted with solid lines.

**Figure 3 genes-14-01170-f003:**
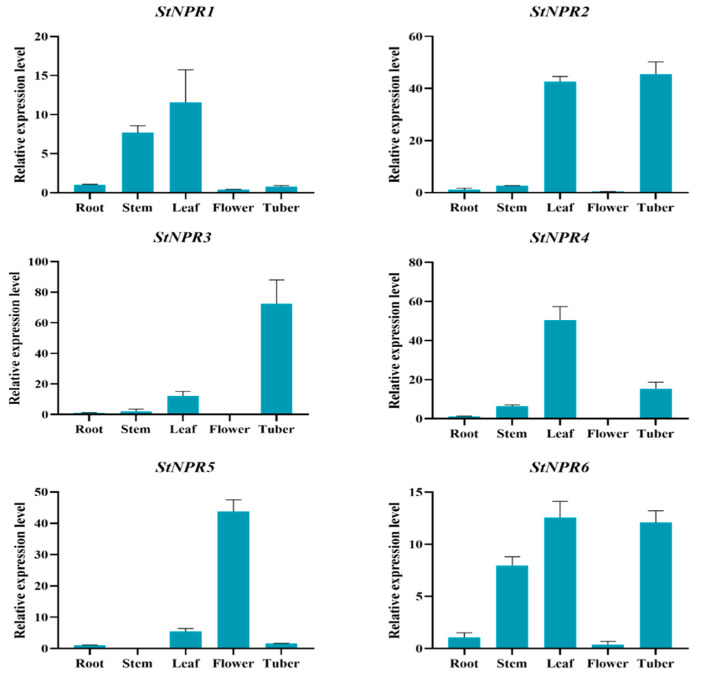
Real-time quantitative PCR analysis of *StNPR1*-like genes’ expression levels in different tissues (root, stem, leaf, tuber, and flower) of potato.

**Figure 4 genes-14-01170-f004:**
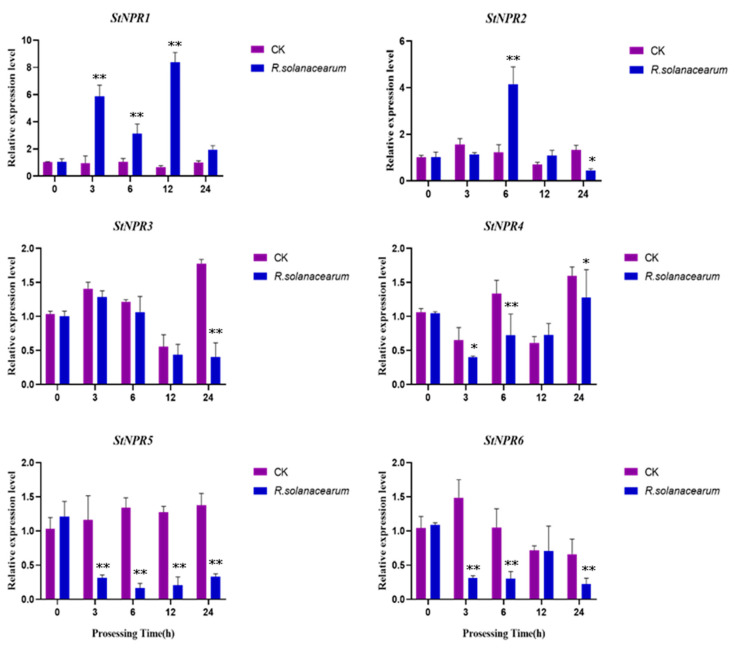
Real-time quantitative PCR (qRT PCR) to validate the relative expression of *NPR1* homologous genes in potatoes infected with *R. solanacearum*. The *R. solanacearum* bacterial suspension was used to treat potato seedlings at the 8–9-leaf stage and water was used as the control (CK). * indicates significant differences in t-tests (* *p* < 0.05, ** *p* < 0.01).

**Figure 5 genes-14-01170-f005:**
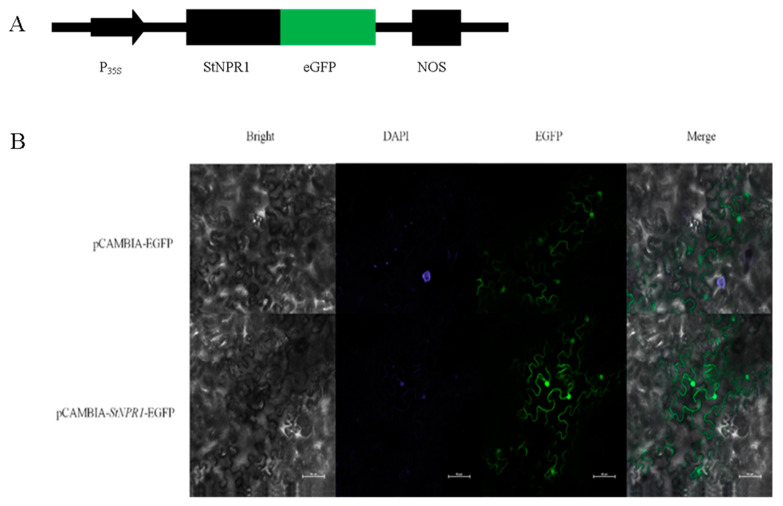
Subcellular localization of StNPR1 protein. (**A**) StNPR1 expression vector construction for tobacco leaf epidermal cells subcellular localization test. P_35S_ indicates CaMV35S promoter, eGFP indicates enhanced green fluorescent protein, and NOS indicates terminator. (**B**) StNPR1 subcellular localization results; bar = 5 mm.

**Figure 6 genes-14-01170-f006:**
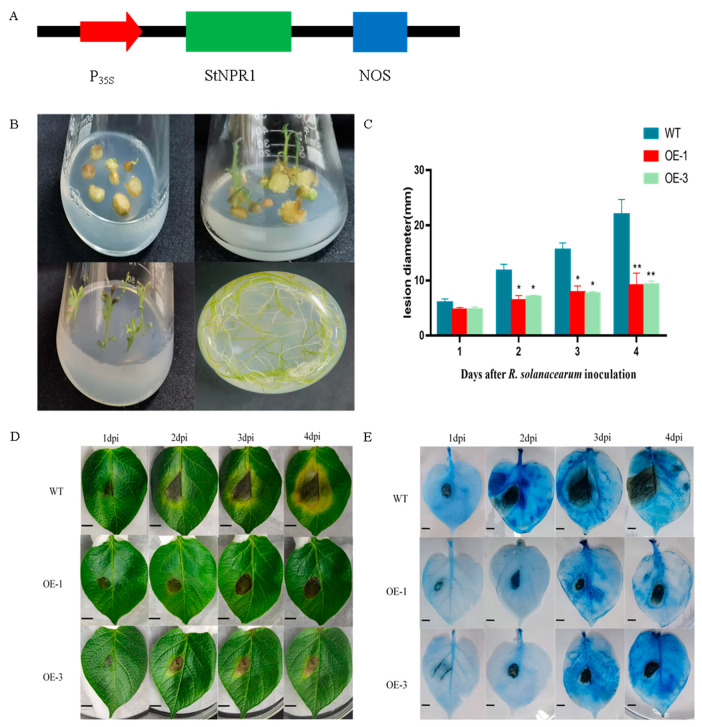
Overexpression of transgenic potato *StNPR1* enhances potato resistance to *R. solanacearum* pathogens. (**A**) Construction of *StNPR1* expression vector. P_35S_ indicates CaMV35S promoter; NOS indicates terminator. (**B**) Obtaining overexpression of *StNPR1* gene in potatoes. (**C**) Comparison of disease spot area of transgenic lines inoculated with *R. solanacearum* and analysis of the proportion of the spot area to the total leaf area. * indicates significant differences in t-tests (* *p* < 0.05, ** *p* < 0.01). (**D**) Symptoms of isolated leaves of transgenic and wild-type lines at different time points after inoculation with *R. solanacearum*. (**E**) Trypan blue staining to detect dead cells. The number of dead cells on leaves of transgenic and wild-type lines at different time points after inoculation with *R. solanacearum*. WT: The isolated leaves of wild-type plants were inoculated with pathogen *R. solanacearum*. OE-1: The isolated leaves of overexpressed 1 were inoculated with pathogen *R. solanacearum*. OE-3: The isolated leaves of overexpressed 3 were inoculated with pathogen *R. solanacearum*.

**Figure 7 genes-14-01170-f007:**
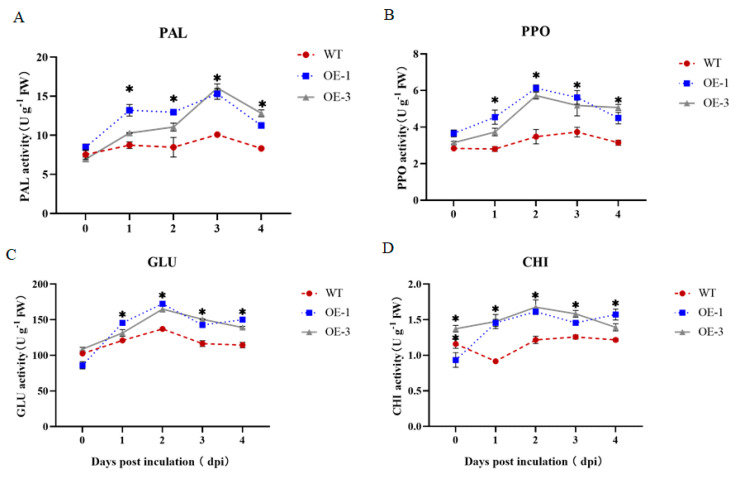
Changes in the activities of four defense enzymes. * indicates significant differences in t-tests (* *p* < 0.05). PAL: Phenylalanine ammonia lyase. PPO: Polyphenol oxidase. Glu: β-1, 3-glucanase. CHI: Chitinase. WT: Inoculation of the isolated leaves of wild-type plants with the pathogen *R. solanacearum* to detect enzyme activity. OE-1: Inoculation of the isolated leaves of overexpressed 1 plants with the pathogen *R. solanacearum* to detect enzyme activity. OE-3: Inoculation of the isolated leaves of overexpressed 3 plants with the pathogen *R. solanacearum* to detect enzyme activity. (**A**) Changes in PAL activity in isolated potato leaves after inoculation with pathogenic bacteria *R. solanacearum*; (**B**) Changes in PPO activity in isolated potato leaves after inoculation with pathogenic bacteria *R. solanacearum*; (**C**) Changes in GLU activity in isolated potato leaves after inoculation with pathogenic bacteria *R. solanacearum*; (**D**) Changes in CHI activity in isolated potato leaves after inoculation with pathogenic bacteria *R. solanacearum*.

**Figure 8 genes-14-01170-f008:**
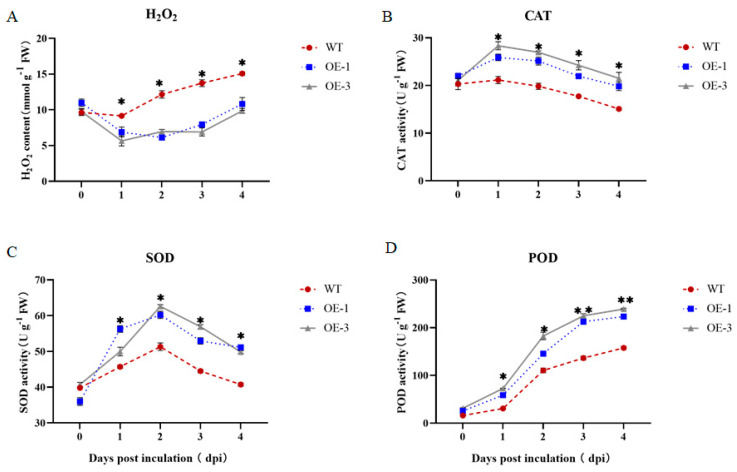
Changes in ROS and antioxidant enzyme activities. * indicates significant differences in t-tests (* *p* < 0.05, ** *p* < 0.01). H_2_O_2_: Hydrogen peroxide. CAT: Catalase. SOD: Superoxide dismutase. POD: Peroxidase. WT: Inoculation of the isolated leaves of wild-type plants with the pathogen *R. solanacearum* to detect enzyme activity. OE-1: Inoculation of the isolated leaves of overexpressed 1 plants with the pathogen *R. solanacearum* to detect enzyme activity. OE-3: Inoculation of the isolated leaves of overexpressed 3 plants with the pathogen *R. solanacearum* to detect enzyme activity. (**A**) Changes in H_2_O_2_ activity in isolated potato leaves after inoculation with pathogenic bacteria *R. solanacearum*; (**B**) Changes in CAT activity in isolated potato leaves after inoculation with pathogenic bacteria *R. solanacearum*; (**C**) Changes in SOD activity in isolated potato leaves after inoculation with pathogenic bacteria *R. solanacearum*; (**D**) Changes in POD activity in isolated potato leaves after inoculation with pathogenic bacteria *R. solanacearum*.

**Figure 9 genes-14-01170-f009:**
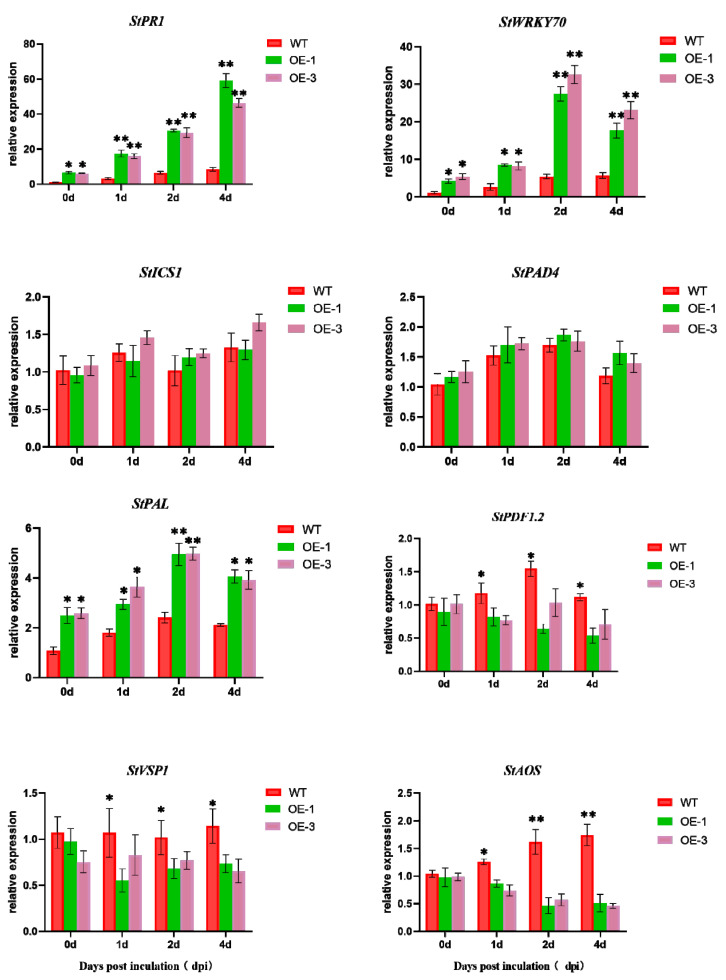
Effect of overexpression of StNPR1 gene on the expression of SA- and JA-defense-related genes. * indicates significant differences in *t*-tests (* *p* < 0.05, ** *p* < 0.01).

## Data Availability

All data is included in this paper and Appendix A.

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
