# Peer review of "Genome-Wide Identification of the NPR1-like Gene Family in Solanum tuberosum and Functional Characterization of StNPR1 in Resistance to Ralstonia solanacearum"

_genes, 2023, doi:10.3390/genes14061170_

Round 1
Reviewer 1 Report
The paper submitted for review does not meet the requirements and standards of a reputable scientific journal. The work is written extremely carelessly. The language is not always understandable. The manuscript requires linguistic revision before resubmission.
There are numerous broken or completely incomprehensible sentences.
The form of preparation of the manuscript makes it impossible to review the manuscript.
After reading the introduction and materials, I almost stopped critical reading.
The reader focuses on the errors instead of the results.
The methodological descriptions of the construction of transgenic plants are too laconic and should be improved. Based on these, repeating the experiments described in this manuscript is impossible. The same applies to the description of the qPCR reaction and the enzymatic activity measurements.
There are numerous spelling errors, missing spaces, or their presence in inappropriate places.
Inconsistent spelling of Latin names of plants and bacteria and gene names that are written in italics or not.
Inappropriate use of terms relating to the potato plant, which is not a cereal plant and, in agriculture, is cultivated as an annual rather than a perennial crop.
In the case of transgenic plants, we do not tend to use the term strain, which is specific to bacteria.
Detailed comments
Abstract
In line 9, the space is missing after the parenthesis.
In lines 11-12, the authors state that Potato is "a crucial grain crop…". However, the Potato is a root crop and not a grain crop.
Shouldn't it be a non-grain food crop or a non-cereal food crop?
In line 22, there is an unnecessary space between the parenthesis and the coma.
Lane 23 and 27, 98, 292, 304, 314, 316 and many other places throughout the manuscript - Can the term "strain" be used for plants? It is used, rather, for bacteria. I suggest replacing it with the term transgenic plant with overexpression of NPR1 gene or overexpressor line or variety.
Introduction
Line 34 - I consider the statement that "Potato (Solanum tuberosum) is a perennial herbaceous plant" unfortunate as potatoes are herbaceous perennials only in nature; however, in modern-day farming, Potato is cultivated as an annual species grown and harvested every year. As the publication concerns the Potato as a crop and the problems during its production, I propose to modify this sentence by, for example, removing the terms "perennial" and "herbaceous".
In line 53, there is no space between the parenthesis and the dot. ").At"
In lines 57-65, the authors describe the six NPR1 genes in A.thaliana, their roles and their division into groups but do not explain on what basis these conclusions are drawn. One can only guess that these are the outcomes of a phylogenetic analysis carried out by other researchers, who identified three phylogenetic groups. A citation is missing, e.g. to the paper by Backer et al. 2019 Front. Plant Sci., 13 February 2019
Sec. Plant-Pathogen Interactions Volume 10 - 2019 | https://doi.org/10.3389/fpls.2019.00102, which precisely addresses these analyses or references to other studies in which this information is included. The description regarding phylogenetic analyses should be added before the information provided by the authors in lines 57-65.
In lane 65, there is an unnecessary space after the parenthesis and citation Hepworthrt al. ( Hepworth et al. 2005.
The spelling of Arabidopsis thaliana should be standardised throughout the manuscript. Italics should be used consistently, the generic name in capital letters and the species name in small letters.
Similarly, the spelling is inconsistent for Nicothiana benthiana.
The Latin name of the bacterium Agrobacterium tumefaciens should be in italics.
Materials and Methods
Line 106 - space is missing ).A
Line 107 – which method was used for phylogenetic analysis? There is no information if the bootstrap analysis was performed.
Line 108 – there is no reference or web page for MEGA 7.0 software.
Line 115 – there is no superscript and no spaces "the concentration 1×108cfu/mL (A660= 0.2)"
Absorbance at 660 0r 600?
Line 119 – font size is too big "-80°C refrigerator for later use."
Line 127 – insert space after actine "Actingene". Why does the Actin gene start from the capital letter?
Replace "GenBank login number:" with GenBank accession number:
For example, line 128 - rewrite and precise the sentence ", and the reaction System was Mx-3000p-Real-Time PCR System".
Line 137 Agrobacterium tumefaciens LBA4404 – the Latin name should be in italic
Line 138 – correct the spelling of the name Nicotiana Benthamiana – and the Latin name should be in italics
Line 146 - correct the spelling of the name Agrobacterium LBA4404 and add the species name
Lines 148-149 The sentence is not clear. It should be rewritten. It is not known what the author had in mind. Use spectrophotometer at wavelength of 600 nm to measure the potato chips with the treatment of RS solution with UV absorption value of 0.5(OD600=0.5) for 7min.
The bacteria solution had an optical density of 0.5 (OD600=0.5), and potatoes inoculated with bacteria were irradiated with UV for 7 min.
Line 149 chips? or potato tuber slices?
Line 151 – add r into the "mictopotato"
Line 152 - The abbreviation of antibiotics should be explained. kan 50mg/L and cef 250mg/L. The name of the differentiation medium should be indicated.
This sentence should be rewritten; it is not clear "Then transfer the mictopotato to the differential medium containing kan 50mg/L and cef 250mg/L in vitro".
153-154. The abbreviation CCC should be explained.
in vitro is usually written in italics. Please correct the spelling throughout the manuscript
line 160 "Potato leaves were cultured in vitro" or Potato plants were grown/cultured in in vitro conditionsLine 161 - This sentence looks like directly copied from protocol "Inject RS solution (OD600=0.8) into the isolated leaves to spread the bacterial solution in the leaves". The manuscript needs linguistic revision.
Line 162 –"The pathological symptoms" should be replaced with "the disease symptoms"
Line 163 –Clarify what "culture 1, 2, 3, and 4d "means.
Line 164 - There is no trypan blue staining method description, or an appropriate reference should be provided. Lines 170-173 - It should be specified for which specific genes the expression level was tested using the qPCR method. There is no reference to the primers used. Lines 170-169 There is no description of the method for determining the activity of enzymes: β-1, 3-glucanase (Glu), chitinase (CHI), phenylalanine ammonia 170 lyase (PAL) and polyphenol oxidase (PPO), superoxide dismutase (SOD), peroxidase (POD), catalase (CAT) and the appropriate references should be provided. Lines 175-177 There is no information about specific statistical methods and software used. ResultsLine 180 Rewrite the sentence. This sentence is incomprehensible. "After the screening (of what?) and identification (of what?),
we identified 6 NPR1 family members were identified in potato, named StNPR1-6 (Table S2).
Figure 1 Explain what the different colours on the tree mean. For example, what do the red triangles mean? You should also refer to these in the text in lines 191-202. Lines 209-231 and legends of Figures 1 and 2 It is worth giving the names of plants from which the sequences were analysed. What do the abbreviations At, Os and Nt mean?Lines 251-252 This sentence should be moved to the discussion paragraph.
Line 254
Add "expression level" after StNPR1-like genes in the description of Figure 3. Real-time quantitative PCR analysis of StNPR1-like genes in different tissues (root, stem, leaf, tuber, flower) of Potato.
Figure 4
What does the shortcut CK means?
The Shortcut RS should be replaced by R. solanacearum throughout the manuscript
There is no information that the expression of individual genes differed depending on the time from inoculation.
Lines 256 – 268. Paragraph 3.4. Expression analysis of StNPR1-like genes under biotic stress The obtained results have not been analysed in detail. This paragraph should be significantly expanded, especially since it is the primary goal of this work. For example, at 12 hours, no statistically significant differences were observed in the level of expression of the three genes NPR2,4 and 6. The level of expression of the NPR3 gene differed only in 24h, while for the NPR1 gene differences were observed between 3 and 6h after inoculation.
Lines 283-304 - The explanation of the shortcut OE-1 and OE-3 is missing both in the manuscript and the caption of Figure 6 and Figure 7.
Line 288 - What do OE-1 and OE-3 lesions mean?
Under Figures 7 and 8, the decipherment of the abbreviations of the enzyme names should be provided.
Lines 342-345 and 350-351
The following sentences," We selected four genes associated with the SA defense response, including the PR1 downstream transcription factor WRKY70, the disease-course-associated gene PR1, the isochorismate synthase gene ICS1, and the Phytoalexin deficient 4 gene PDA4 (Lee et al. 1995; Li et al. 2004; Morris et al. 2000; Wildermuth et al. 2001)" and "Transcriptional analysis was also performed on JA biosynthesis gene AOS, JA response gene VSP1, and the marker gene PDF1.2 which is as- sociated to JA defense response, (Petersen et al. 2000; Rojo et al. 1999; Von et al. 2002), describing the selection of gens used for transcriptional analysis of SA and JA signalling pathway-related genes should be moved to the Methods section.
Author Response
Dear Editor and Reviewers
我们代表我的合著者,非常感谢您给我们机会修改我们的手稿,我们非常感谢编辑和审稿人对我们题为“Solanum tuberosum中NPR1样基因家族的全基因组鉴定和StNPR1在抗Ralstonia solanacearum中的功能表征”的手稿的积极和建设性的意见和建议。(手稿编号:genes-2387179)
我们仔细研究了审稿人的意见,并对论文中的红色标记进行了修改。我们已尽力根据意见修改我们的稿件。我们让英语语言专业人员验证了手稿,验证报告在最后一页。随函附上订正本,谨提交该订正本供你审议。

Reviewer 2 Report
The manuscript “Genome‑wide identification of the NPR1‑like gene family in Solanum tuberosum and functional characterization of StNPR1 in resistance to Ralstonia solanacearum” by He et al sent for publication to Genes deals with important topic such as NPR-1 family gene characterization and their implication to important potato disease such as bacterial wilt. The manuscript will be of interest to the scientific community working on the topic crop disease resistance. The present work deserve to be published after addressing some comments listed below:
Remarks:
L2 There is typos in the title: “Identifcation” should be “Identificaion”.
The english language should be verified by native speaking person.
All latin names should be in italics and the gene names too. (some examples: L137 Agrobacterium tumefaciens, L139 Nicotiana b. etc…)
Abstract:
L23: Th eauthors wrote “… established potato StNPR1 overexpression strains…”. The term strain used for plants is not appropriate.The authors should use clone/ line. Strains is more related to bacteria.
Introduction:
The introduction is well written and point the main problems related to the topic.
Materials and methods:
This part needs improvement. L127: Actingene should be “actin gene” in italic the gene name and remove the repeated gene name.
In section 2.3 the authors describe the construction of the binary vector pBI121 using homologous recombination (HR). Are they sure that they used HR to clone the gene? Some more explanation in details is needed.
L148-150: I didn’t understand what the authors mean with that sentence. It needed to be more clear and in details. As well all antibiotics names should be given in full, not “kan”, “cef”.
L168: they should mention which genes they used for transcriptional analysis.
L164: The authors used trypan blue staining but in L303 they use taipan. Please correct.
Results and analysis:
This part is somehow well written but some improvement is needed. For example, L 180-181 is not clear.
Discussion:
This section is well written and supported by results but I have a remark: L380 the authors wrote: ” RS is one of the major diseases of potato and is the most serious bacterial disease which seriously threatens the yield and quality of potato…”
RS is the pathogen not the disease, the disease is bacterial wilt. The authors need to rewrite this.
Overall, the manuscript deserved to be published after addressing the comments and improve the manuscript.
The english language should be verified by native speaking person.
Author Response
Dear Editor and Reviewers
On behalf of my co-authors, we thank you very much for giving us an opportunity to revise our manuscript, we appreciate editor and reviewers very much for their positive and constructive comments and suggestions on our manuscript entitled “Genome‑wide identification of the NPR1‑like gene family in Solanum tuberosum and functional characterization of StNPR1 in resistance to Ralstonia solanacearum”. (Manuscript ID: genes-2387179)
We have studied reviewer’s comments carefully and have made revision which marked in red in the paper. We have tried our best to revise our manuscript according to the comments. We had the manuscript validated by an English language professional and the validation report is on the last page. Attached please find the revised version, which we would like to submit for your kind consideration.

Reviewer 3 Report
In this paper the authors report on their extensive analyses of StNPR1-like gene expression in potato subject to Ralstonia solanacearum infection. Both resistance and effects on some defense genes are described. My main reservation about publication is the fact that effects which are described are relatively modest (about 50% in Figures 6-8) and actually may be indirect, so biologically significant conclusions cannot be drawn. as the last sentence recognizes ("StNPR1 could be"). Nevertheless, the data represent a lot of extensive analyses and should be useful to other researchers in the field.
Overall the paper is somewhat wordy with some awkward language structure. It would benefit from additional editorial review.
Author Response

(The authors gave the same response as above.)

Round 2
Reviewer 1 Report
The authors have corrected the manuscript and complied with all comments and remarks
The article is suitable for publication